# Are Natural Compounds a Promising Alternative to Synthetic Cross-Linking Agents in the Preparation of Hydrogels?

**DOI:** 10.3390/pharmaceutics15010253

**Published:** 2023-01-11

**Authors:** Paulina Sapuła, Katarzyna Bialik-Wąs, Katarzyna Malarz

**Affiliations:** 1Department of Organic Chemistry and Technology, Faculty of Chemical Engineering and Technology, Cracow University of Technology, 24 Warszawska St., 31-155 Cracow, Poland; 2A. Chelkowski Institute of Physics, Faculty of Science and Technology, University of Silesia in Katowice, 75 Pułku Piechoty 1A, 41-500 Chorzow, Poland

**Keywords:** natural and synthetic cross-linking agents, hydrogels, biocompatibility, environmental protection

## Abstract

The main aim of this review is to assess the potential use of natural cross-linking agents, such as genipin, citric acid, tannic acid, epigallocatechin gallate, and vanillin in preparing chemically cross-linked hydrogels for the biomedical, pharmaceutical, and cosmetic industries. Chemical cross-linking is one of the most important methods that is commonly used to form mechanically strong hydrogels based on biopolymers, such as alginates, chitosan, hyaluronic acid, collagen, gelatin, and fibroin. Moreover, the properties of natural cross-linking agents and their advantages and disadvantages are compared relative to their commonly known synthetic cross-linking counterparts. Nowadays, advanced technologies can facilitate the acquisition of high-purity biomaterials from unreacted components with no additional purification steps. However, while planning and designing a chemical process, energy and water consumption should be limited in order to reduce the risks associated with global warming. However, many synthetic cross-linking agents, such as *N*,*N*′-methylenebisacrylamide, ethylene glycol dimethacrylate, poly (ethylene glycol) diacrylates, epichlorohydrin, and glutaraldehyde, are harmful to both humans and the environment. One solution to this problem could be the use of bio-cross-linking agents obtained from natural resources, which would eliminate their toxic effects and ensure the safety for humans and the environment.

## 1. Introduction

Hydrogels constitute hydrophilic three-dimensional networks, which are obtained through the cross-linking reactions of synthetic and/or natural polymers. They are primarily characterised by their ability to absorb water from 10–20% (an arbitrary lower limit) up to thousands of times their dry weight [1,2,3]. Polymers of a natural origin are becoming increasingly used as raw materials to prepare the hydrogels. This is due to their low cost and the lack of a negative impact on the environment, as well as the biofunctionality of the derivative products. The most frequently used groups of natural polymers are polysaccharides, e.g., alginate, chitosan, hyaluronic acid, and proteins (collagen, gelatin, fibrin). Generally, these polymers enable the production of hydrogels that offer desirable properties such as biocompatibility, biodegradability, and non-cytotoxicity. Given their various positive features, hydrogel materials have been widely used in the pharmaceutical and medical industries, e.g., in wound healing dressings, controlled drug or vaccine delivery systems, soft tissue engineering, tissue implants, hygiene products, and in the production of contact lenses (Figure 1) [2,4,5,6,7,8,9,10].

Hydrogel materials, including the matrices that have been modified using active substances, can be prepared using various methods, such as chemical or physical cross-linking, under the influence of microwave, UV, and gamma radiation [3,11,12,13,14,15]. However, the products that are characterised by an extended-release time of the active substance, e.g., up to seven days, the chemical cross-linking of the polymer matrix should be used, which often requires the use of synthetic cross-linking agents. In some cases, the physical cross-linking methods are insufficient and the active substance is repeatedly released too rapidly, as was confirmed by the latest reports [13]. Currently, material engineering is primarily based on synthetic cross-linking agents, such as *N*,*N*′-methylenebisacrylamide, ethylene glycol dimethacrylate, poly (ethylene glycol) diacrylates, epichlorohydrin, and glutaraldehyde. The use of such compounds carries a huge risk of toxic effects if the cleaning process is not performed properly [16,17]. One solution to this problem could be using bio-cross-linking agents that are obtained from natural resources, which are considered to be safer, free of toxic effects, and do not require a purification stage, which is also associated with the development of more ecological methods for obtaining the hydrogels. It should be emphasised that using high-purity materials from unreacted components that do not require additional purification steps is expensive and consumes energy, water, etc. Therefore, the main aim of this review is to present new natural bio-cross-linkers that could be used as alternatives to their synthetic equivalents.

## 2. Biopolymers Used to Prepare Hydrogels

In the last few decades, natural polymers have become widely used as macromolecular compounds to prepare hydrogels for medical, biosensing, and biomedical applications. Based on their chemical structure, it is possible to divide them into three basic groups: polysaccharides, proteins and polypeptides, and polynucleotides. The classification of the natural biopolymers that are commonly used to prepare hydrogels is presented in Figure 2.

### 2.1. Polysaccharides

Polysaccharides belong to the group of complex carbohydrates that are composed of more than ten monosaccharides. These monosaccharides are bonded by glycosidic linkages in linear or branched chains that have a molecular weight of tens of thousands or even millions. The sources of polysaccharides include both plant organisms (alginates, starch, cellulose) and animal organisms (chitin, hyaluronic acid, glycogen) [18,19,20]. Moreover, polysaccharides are often used to prepare hydrogels due to their non-toxicity, biocompatibility, biodegradability, and susceptibility to solubilisation [21]. As a result, the hydrogels that are obtained using polysaccharides have been widely used in the fields of biotechnology, medicine, and biomedicine, e.g., for wound dressings, tissue engineering, drug delivery systems, orthopaedic surgery, and ophthalmic procedures. Despite their many positive aspects, the hydrogels that are based on polysaccharides have some disadvantages, such as poor mechanical properties and an instability in the body’s fluid system. That is why it is important to perform the correct cross-linking reaction or to modify them with additional components (e.g., synthetic polymers, inorganic or organic nanoparticles) in order to strengthen the structure of the hydrogel matrix while still preserving their biological properties [22,23].

#### 2.1.1. Alginates

Alginates are polysaccharides that are composed of α-D-mannuronic acid and β-L-guluronic acid, which are obtained through the extraction of seaweed, and their molecular weight is in the range of 10–1000 kDa. The main characteristics of alginates are their non-toxicity and biocompatibility. Moreover, they have a non-immunogenic effect, do not cause inflammatory reactions, and have mucoadhesive properties. Alginate is distinguished by its high water absorption; it can absorb between 200 and 300 times more water than its own weight. Furthermore, alginate-based hydrogels are characterised by their good mechanical strength and a favourable degradation rate. Interestingly, alginate-based biomaterials can take different forms, such as particles, fibres, scaffolds, sponges, or membranes. For these reasons, they have been used, *inter alia*, in tissue engineering, drug delivery systems, and as a component in wound dressings [6,24,25,26].

#### 2.1.2. Chitosan

Chitosan is one of the most important chitin derivatives and can be obtained thought chemical or enzymatic deacetylation, whereas the natural sources of chitin include the shells of marine crustaceans and the cell walls of some types of fungi. Chitosan belongs to the linear polysaccharides. It is composed of β-(1,4)-D-glucosamine and *N*-acetyl-D-glucosamine, i.e., the deacetylated part and the acetylated part, respectively. The main features of chitosan are its bioactivity, biocompatibility, non-toxicity, and biodegradability. Moreover, it undergoes to modify it chemically or physically, which minimises its disadvantages such as its limited solubility in certain solvents and its poor mechanical properties. Chitosan is used to prepare biomedical materials such as dressings, orthopaedic materials, and the hydrogels that have a controlled drug delivery [6,27,28,29,30].

#### 2.1.3. Cellulose

Cellulose is the most abundant biopolymer on Earth and it is the main structural component of plant cell walls. The basic sources of cellulose are wood, cotton, straw, and flax. The cellulose material consists of β-D-glucose (anhydroglucose) units that are linked by β-1,4-glycosidic bonds, and the substance is characterised by a semi-crystalline structure with both crystal and amorphous domains. This polymer has the required properties for biomaterials: biocompatibility, bioactivity, biodegradability, and non-toxicity. Moreover, cellulose is also characterised by its low reactivity and solubility. To improve these properties, the surface of the molecule is often modified. A chemical modification of the polymer surface limits the hydrogen interactions between two molecules, among other things. The methods that are used for this type of modification include physical interactions or the adsorption of molecules on the cellulose surface, as well as in the covalent bonds between the polymer and the grafting agent. Cellulose has also been used to obtain biomaterials that are produced to treat long-healing wounds [31,32,33,34,35,36].

#### 2.1.4. Hyaluronic Acid

Hyaluronic acid belongs to the group of polysaccharides that are called glycosaminoglycans and it is composed of two repeating units—D-glucuronic acid and *N*-acetyl-D-glucosamine. The largest amounts are primarily found in the skin, synovial fluid, and the eye’s vitreous body. Hyaluronic acid is characterised by its biocompatibility, non-immunogenicity, and viscoelastic properties. By chemically modifying the compound, it is possible to improve its physical properties, such as its solubility and degradation rate. An example of this type of modification is the esterification process of hyaluronic acid or the cross-linking process. The compound is used in orthopaedic surgery, ophthalmic procedures, the production of dressing materials, and plastic surgery [24,25,37,38,39].

### 2.2. Proteins

Proteins are macromolecular compounds that belong to the group of natural polymers that contain more than 20 different amino acids, connected by peptide bonds and that adopt a complex three-dimensional structure. Depending on the shape of the molecule, the proteins are divided into spherical globular proteins (e.g., haemoglobin, insulin, albumin) and elongated fibrillar proteins (e.g., collagen, gelatin, fibroin). The protein-based hydrogels, which are prepared using collagen, gelatin, or fibroin, are mainly distinguished by their excellent biocompatibility and bioactivity. Moreover, this type of material is commonly used in the medical industry. However, it has some disadvantages, including mechanical instability or, in the case of fibrin gels, rapid degradation. Therefore, it is better to blend them with natural and synthetic polymers in order to increase their cross-link density and obtain more stable hydrogels [40,41,42,43,44,45,46].

#### 2.2.1. Collagen

Collagen is the main protein that forms the intercellular matrix structure and constitutes one-third of all of the proteins in the human body. This protein is found, *inter alia*, in the bones, cartilage, tendons, and skin where it has a structural function. Type I collagen accounts for between 80% and 99% of all of the collagen in the human body and belongs to the group of fibrillary collagens that form highly organised fibres and fibrils, which act as building materials. Collagen is characterised by its excellent biological properties, biocompatibility, non-toxicity, and biodegradability. Moreover, it has a low antigenicity and a weak inflammatory response. Its main disadvantage is that it has insufficient mechanical and thrombogenic properties due to the degradation process. Collagen is used in ophthalmology, surgery, the production of dressing materials and sustained drug delivery systems, and in tissue engineering [6,24,47,48,49].

#### 2.2.2. Gelatin

Gelatin is a fibrous protein that consists of repeating units of three amino acids: glycine, proline, and hydroxyproline. It is obtained through the process of the partial acid or alkaline hydrolysis of collagen, which generates type A or type B gelatin, respectively. The sources of gelatin are cattle and pork skin, bovine and pork bones, and fish, among others. Gelatin is characterised by its biocompatibility, non-immunogenicity, low antigenicity, and biodegradability. It is highly absorbent and can absorb between five and ten times more water than its initial weight. However, the main disadvantages of gelatin are its low thermal stability and insufficient mechanical properties. Gelatin is used, *inter alia*, as a dressing material, in the production of soft and hard capsules, drug delivery systems, and in tissue engineering [50,51,52,53,54,55].

#### 2.2.3. Fibroin

Fibroin is a protein that is the main component of the silk fibres that are produced by spiders and some insects. It is primarily composed of protein domains, which include the amino acid residues of lysine, alanine, and serine. Interestingly, it has very good mechanical properties and is high flexible. Moreover, fibroin is characterised by its biocompatibility, biodegradability, low inflammatory potential, and low immunogenicity. Fibroin-based biomaterials can take the form of hydrogels, nanofibers, microspheres, membranes, sponges, films, and scaffolds, which offer a variety of uses such as in tissue engineering, surgical materials, and controlled drug delivery systems [46,56,57,58,59,60].

The selected properties of the natural biopolymers that are used to prepare hydrogels are presented in Figure 3.

## 3. Cross-Linking Reaction

The cross-linking process is based on the reaction of the functional groups of the polymer chain (–OH, –COOH, –NH_2_) with cross-linking agents, which leads to the formation of new bonds. As a result, the polymer material is cross-linked. Hydrogels constitute a network of linked polymer chains that have a three-dimensional structure. Depending on the forces that are present during the process, physical and chemical cross-linking can be distinguished. The cross-linking reaction enables the formation of covalent, hydrogen, and ionic bonds as well as van der Waals interactions between the polymer chains. In addition, the cross-linking reaction is used to modify various characteristics of polymer materials, such as their thermal, mechanical, and physicochemical properties [24,61,62].

### 3.1. Chemical Cross-Linking Reaction

The chemical cross-linking reaction produces mechanically strong hydrogels that can absorb water and other fluids in a reversible manner. In addition, such materials are characterised by their stability under physiological conditions and an extended degradation time. One of the methods for cross-linking polymeric materials is using classical organic reactions between the functional groups of the polymer being used. Examples of such reactions include the Huisgen cycloaddition, the Diels–Alder reaction, the Michael addition, and the Schiff base reaction (Figure 4) [3,11,63,64].

#### 3.1.1. Huisgen Cycloaddition

The Huisgen reaction is a 1,3-dipolar cycloaddition that occurs between an azide and a terminal alkyne. The process is most often catalysed using copper or ruthenium salt, and its product is a 1,4-disubstituted triazole or a 1,5-disubstituted triazole. The copper catalyst is characterised by its high cytotoxicity and relatively high cost. Moreover, the catalyst is only slightly recyclable and requires the use of organic solvents. The Huisgen cycloaddition is a highly regioselective process that obtains an excess of one of the structural isomers of the product. Moreover, the reaction is highly efficient and can be conducted at room temperature and in a mild environment, e.g., in water. The cycloaddition reaction is used in the synthesis of the hydrogels that are based on chitosan, hyaluronic acid, and cellulose [65,66,67,68,69,70,71,72].

#### 3.1.2. Diels–Alder Reaction

The Diels–Alder reaction is a method for forming C–C bonds by the selective [4 + 1] cycloaddition between a diene and a dienophile (e.g., a substituted alkene). Conducting the reaction in an aqueous environment offers a significant acceleration of its course, although the total duration under the physiological conditions remains long—up to several hours. Cross-linking a hydrogel using the thermoreversible cycloaddition reaction occurs in one step, does not require the addition of an initiator or catalyst, and no by-products are formed during its course. Moreover, the hydrogels that are obtained using the Diels–Alder reaction are slightly degraded due to the slow rate of the retro-Diels–Alder reaction (the reverse process) at body temperature. The Diels–Alder reaction is used in the synthesis of the hydrogels that are based on pectin, chitosan, starch, and chitin [73,74,75,76,77,78].

#### 3.1.3. Michael Addition

The Michael reaction is a method that is used to form a carbon–carbon bond via a nucleophilic addition that involves attaching a carbanion or nucleophile (amines, thiols) to an activated α,β-unsaturated carbonyl polymer. The two basic variants of the addition are the aza-Michael and thio-Michael reactions, in which a carbon-nitrogen or carbon–sulfur bond is formed, respectively. The reaction proceeds under a physiological temperature and pH conditions and does not require cytotoxic initiators, free radicals, or UV radiation. The Michael addition is characterised by its high regioselectivity, high efficiency, and relatively short reaction time. Furthermore, it is an irreversible process and no by-products are formed. The resultant hydrogels have a good biocompatibility and have a homogeneous polymer network. The Michael addition is used in the synthesis of the hydrogels that are based on gelatin, chitosan, collagen, and hyaluronic acid [79,80,81,82,83,84,85,86].

#### 3.1.4. Schiff Base Reaction

The hydrogels that are obtained as a result of the Schiff base reaction are characterised by the presence of a covalent imine bond in their structure, which is formed by the coupling of the aldehyde and amine groups with the polymer chains. The mechanism of the process is based on a nucleophilic attack of the nitrogen atom in the amino group on the electrophilic carbon atom in the aldehyde or ketone molecule. The Schiff base reaction is a reversible process and occurs in an aqueous environment under physiological conditions. The reaction offers a short duration time, a high chemical selectivity, and non-toxic products. The hydrogels that are obtained using this method are self-healing, biocompatible, and sensitive to pH changes. The Schiff base reaction can be used in the synthesis of the hydrogels that are based on starch, chitosan, gelatin, and alginate [79,87,88,89,90,91,92].

#### 3.1.5. Emulsification Method

The emulsification method consists of introducing a hydrogel precursor into a hydrophobic environment and constantly stirring the newly-created nanoemulsion system. When the water phase breaks into smaller droplets, the hydrogel microspheres are formed. These can be cross-linked using other methods, such as photopolymerisation or ionic cross-linking methods. The nanoemulsions can be created using low- or high-energy methods. The low-energy method is based on a phase inversion and occurs under atmospheric pressure, while the high-energy microfluidisation method requires high pressure. The size of the microspheres can be modified by changing the viscosity of the nanoemulsion and the intensity of the mixing. The main disadvantages of the emulsification method are the lack of homogeneity in the microspheres and the possibility that they can change their shape. The emulsification method is used in the synthesis of three hydrogels that are based on chitosan and starch [7,93,94,95,96,97,98].

## 4. Cross-Linking Agents from Synthetic and Natural Sources

Inducing a cross-linking reaction by introducing a cross-linking agent into a system is a commonly used method for obtaining the hydrogel materials that are based on biopolymers. The traditionally used chemical cross-linkers include glutaraldehyde, glyoxal, poly (ethylene glycol), dialdehyde starch, and 1-ethyl-3-(3-dimethylaminopropyl) carbodiimide. These compounds enable the effective cross-linking of polymers and positively affect the properties of the final materials. Unfortunately, most chemical cross-linkers are highly toxic to the cells in a human body. Therefore, these biomaterials must undergo complex purification processes to remove any unreacted toxic compounds. The side effects of using these compounds include the inflammation and irritation of the skin and mucous membranes, kidney diseases, and headaches. An alternative that is increasingly gaining importance is using cross-linking agents of a natural origin, such as genipin, citric acid, tannic acid, epigallocatechin gallate, and vanillin. The natural compounds that are used in the cross-linking reaction are characterised by a much lower cytotoxicity and, in some cases, they are entirely non-cytotoxic. Natural cross-linkers create stable chemically cross-linked hydrogel materials with a high biocompatibility [99,100,101,102,103].

The summary presented in Table 1 compares the advantages and disadvantages between the cross-linking agents of natural and synthetic origins. Further in this section, the compounds representing both groups of cross-linking agents are characterised.

### 4.1. Synthetic Cross-Linking Agents

#### 4.1.1. Glutaraldehyde

Among all of the compounds that are used, glutaraldehyde is used as a cross-linking agent most often because it has the highest cross-linking efficiency compared to the other aldehydes. Glutaraldehyde is characterised by its excellent solubility in water and organic solvents and it is available as an oily liquid in a colour gradient from colourless to a pale straw. The main advantages for its wide use in cross-linking reactions are its easy availability, low cost, and high reactivity. Moreover, glutaraldehyde can react with the amine and hydroxyl functional groups that are found in a wide variety of proteins and polysaccharides. It was found that the biomedical materials that are obtained as a result of their use have a high tensile strength and good mechanical properties. The presence of the aldehyde groups in the structure of the final product causes a toxic effect on cells and a manifestation of its cytotoxicity is the presence of inflammations in the body. The diseases that are caused from contact with glutaraldehyde include skin or eye irritation, contact dermatitis, headache, irritant contact dermatitis, chronic dermatitis, and systemic sensitisation. For this reason, it is necessary to use special purification processes for biomaterials. One example of this process is washing the hydrogels with amino acid solutions or fluids that contain the free amino groups. The research efforts determined that the limit for its concentration is 8% *w*/*v*, which allows for the elimination of the toxic effect of glutaraldehyde after a purification process [22,99,110,111,112].

Cross-linking collagen using glutaraldehyde occurs as a result of the reaction of the aldehyde group with the ε-amino groups, which are derived from the collagen lysine and hydroxylysine residues. The main intermediate product of the reaction is a Schiff base, which is the substrate for the subsequent intermediate reactions. The obtained cross-linked protein structure is shown in Figure 5 [113,114].

Martucci et al. [115] described gelatin films that were obtained using cross-linking with glutaraldehyde in which the final content of the cross-linking agent ranged from 0.1 to 0.5 wt%. As a result, the films with a high degree of cross-linking were obtained and the maximum degree of cross-linking reached 97% for the cross-linking agent at a concentration of 2 wt%. Moreover, for these materials, there was a clear correlation between the degree of cross-linking and the creep response. An increase in the degree of cross-linking led to a decrease in the creep response. It was found that the films with a high degree of cross-linking were characterised by a better stiffness than the materials with a lower value of this parameter.

Bigi et al. [111] conducted a cross-linking reaction of a 5% aqueous gelatin solution using glutaraldehyde at a concentration of 0.125 to 2.5 wt%. The cross-linked samples required a double wash with double-distilled water. Based on the mechanical tests, it was found that using even low concentrations of glutaraldehyde significantly stiffened the films and had a particular effect on their stretching properties. The degree of cross-linking of the materials that contained 0.1 to 1 wt% of glutaraldehyde was between 60 to 100%. The highest degree of cross-linking was 98% in the film that contained 1 wt% of the cross-linking agent. The degree of swelling is a parameter that depends on the concentration of a cross-linking agent and it decreases with an increase in the degree of cross-linking in a film. Bigi et al. [111] noted that the swelling ratio was 112% after one hour for a material that contained 1 wt% of glutaraldehyde. In addition, the cross-linking resulted in an increase in the thermal stability of the materials, which was manifested by an increase in the temperature of the gelatin denaturation. As a result of the cross-linking reaction, Bigi et al. obtained a film with a denaturation temperature that increased from 43 °C to about 72 °C. Their study showed that glutaraldehyde was released from the films with the highest content of this substance (2 and 2.5%), and the amount of the released cross-linking agent was about 1 wt% after 24 h.

Talebian et al. [116] obtained films using glutaraldehyde with amounts from 0.01 wt% to 0.2 wt%. Their mechanical tests concluded that, with an increase in the content of the cross-linking agent, the breaking load increased and there was slight elongation at the break. The authors obtained the materials with a significant increase in the breaking load and an extended degradation time, which caused a limited degree of swelling compared to the non-cross-linked material. The highest value of the swelling ratio was obtained for the film that contained 0.06 wt% of the cross-linking agent, and after an hour it was approx. 1300%. The amount of glutaraldehyde that was released from the materials gradually decreased as the content of the cross-linking agent in the film increased from 0.1 wt% to 0.2 wt%. This was similar in the surface structure of the materials, where the surface became smooth as the concentration of the cross-linking agent increased and the number of pores decreased.

Imani et al. [117] described sponges that were obtained using cross-linking with glutaraldehyde solutions at a concentration of 0.5, 1 or 1.5% (*w*/*v*). The cross-linked samples required a triple wash with double-distilled water. It was found that neither the concentration of gelatin nor the cross-linking agent had a significant effect on the size and shape of the pores inside the structure of the sponges. The concentration of glutaraldehyde was important in the material degradation process, and the authors observed that the resistance of the sponges to the degradation increased as the degree of cross-linking increased. The highest degree of swelling was found in the sponge that contained 0.5% (*w*/*v*) of the cross-linking agent and the material was able to absorb 40 times its dry weight.

#### 4.1.2. Glyoxal

Glyoxal belongs to the group of aldehydes that contain two aldehyde functional groups. The presence of two adjacent carbonyl groups is responsible for the high reactivity of this compound. Glyoxal has high water solubility, the vapour pressure of its solution is low, and it is characterised by a much lower cytotoxicity compared to most chemical cross-linking agents. However, when the concentration of glyoxal is more than 5 mM, it may induce oxidative stress in the body, i.e., an imbalance between the free radicals and antioxidants, which can cause a variety of diseases such as hormonal disorders, cardiovascular diseases, or accelerated skin ageing. Unreacted aldehyde molecules can cause toxic effects in the body. Therefore, it is necessary to determine the minimum amount of the cross-linking agent that enables the complete cross-linking of the polymer. When glyoxal comes in contact with the human organism, it exacerbates the development and progression of diabetic nephropathy. Moreover, it can contribute to the development of Alzheimer’s and Parkinson’s diseases [106,118,119,120,121,122]. The cross-linking reaction with glyoxal occurs according to a homobifunctional mechanism, i.e., a one-step reaction. This process might occur as a result of the formation of acetal between the glyoxal aldehyde and hydroxyl groups that are found in the glucosamine residues present in chitosan. The second mechanism of cross-linking is based on the formation of a Schiff base as a result of the reaction between the free amine groups of chitosan and the aldehyde groups of glyoxal (Figure 6) [106,119,123].

Kaczmarek-Szczypczyńska et al. [106] conducted research using a 40 wt% glyoxal solution and a 2% chitosan solution in 0.1 M of acetic acid. The authors were careful not to use more chitosan-reactive monomers as the glyoxal residues can be toxic. The cross-linked materials were washed with distilled water. As a result, a hydrogel was obtained by introducing glyoxal at a concentration of 10% in relation to the polymer to the chitosan solution. The structure of the hydrogel was characterised by the presence of open, interconnected pores. Based on the mechanical tests, the authors determined that the value of the compression modulus was approximately 0.8 kPa, while the percentage of the deformation at the maximum voltage was less than 5%. Furthermore, the degree of swelling after one hour was approximately 400% and there was a decrease in the sample mass due to the degradation process after seven hours of incubation. The blood compatibility tests showed that the haemolysis rate coefficient for the hydrogel material that was not subjected to water rinsing was 63.84%, but after rinsing, the coefficient was reduced to a negative value.

Gupta et al. [124] obtained microspheres cross-linked in acidic conditions with glyoxal by using half the concentration of the polymer solution compared to the previously described test and a cross-linking agent of 2 to 12% (*w*/*w*). The cross-linked microspheres were then washed with distilled water. Using the scanning electron microscopy method, the size of the microspheres ranged from 119 µm for the non-cross-linked material to 31.60 µm for the material that contained 12% (*w*/*w*) of the cross-linking agent. The microspheres were characterised by a smooth morphology compared to the non-cross-linked samples. The cross-linking reaction reduced the shape factor value from 0.922 to 0.795, causing the cross-linked material to be less spherical. The hydrophobic nature of the microspheres increased with an increase in the degree of cross-linking. The swelling ratio reached its maximum after the sample that contained 4% (*w*/*w*) of the cross-linking agent was incubated for 30 min. The value of the coefficient was 200%.

Another study by the same authors [125] used a 2% (*w*/*w*) solution of chitosan in 0.1 M of acetic acid and a 10% (*w*/*w*) glyoxal solution. The content of the chitosan and the glyoxal solution was from 2 to 8% (*w*/*w*). The research confirmed that an increase in the degree of cross-linking resulted in a recipe for hydrogels with a decreased swelling ability. Moreover, an increase in the concentration of the cross-linker solution caused the formation of smaller pores in the structure of the final product. The sample with the highest content of the cross-linking agent was characterised by the maximum value of the porosity of 88.2%, but the lowest value of the void fraction of 0.72 mL/g. The authors observed a significant influence of the cross-linking agent content on the mechanical properties of the material. The penetration pressure of the hydrogels increased from 61 g/cm^2^ for the sample that contained 2% (*w*/*w*) of the cross-linking agent to as much as 202 g/cm^2^ for the sample that contained 8% (*w*/*w*) of the cross-linking agent. The authors also confirmed the porous surface of the hydrogel materials using scanning electron microscopy.

Chang et al. [126] described their preparations of aerogels. In their experiment, a 0.5% (*w*/*v*) solution of chitosan was prepared in a 1 vol% acetic acid solution and glyoxal. The mass ratio of the crosslinking agent relative to water was 1:7, 1:14, and 1:35. After the process, the unreacted cross-linker was removed and replaced with absolute ethanol. The material was obtained by drying it in supercritical conditions, which produced a material that had a large surface with mesopores. The specific surface area of the aerogel, which was stable up to 100 °C, decreased as the concentration of the cross-linking agent increased, and the highest value of this parameter was 707 m^2^/g.

#### 4.1.3. Poly (ethylene glycol) (PEG)

Poly (ethylene glycol) is a polymer that is commonly used in biomedical materials and it belongs to the group of synthetic amphiphilic polyethers. As a commercial product, this compound is available in several variants that differ in their molecular weight. Poly (ethylene glycol) is soluble in both water and organic solvents and it is characterised by its non-toxicity, biocompatibility, limited immunogenicity, and non-antigenicity. Moreover, it is possible to quickly remove PEG from the body when the molecular weight of the compound is below 1000. In addition, poly (ethylene glycol) can be used with various reactive groups. Poly (ethylene glycol) diglycidyl ether (PEGDE) is an example of a PEG derivative with repeating units of ethylene oxide. PEGDE is a cross-linking agent with a flexible polymer chain that is biocompatible and non-toxic. However, tests performed on human cervical cancer cells (HeLa) and a cell line of fibroblasts derived from mice (L929) confirmed a partial cytotoxicity of PEG oligomers and PEG-based monomers. The compounds that exhibit a toxicity to the above-mentioned cells include PEG-1000, PEG-4000, mPEGMA-950, TEG, mPEGMA-500, and mPEGA-480. Using high concentrations of PEG causes notable toxicity in humans, which primarily affects the kidneys. The reported cases include acute renal insufficiency, acute renal necrosis, oliguria, and azotemia. The mechanical properties of these cross-linked hydrogels can be modified by changing the concentration or molecular weight of PEG, which changes the cross-linking density of the material. Therefore, the diffusion of the active substances in the hydrogel can be reduced [107,127,128,129,130,131,132].

For example, the cross-linking of hyaluronic acid is based on its reaction with poly (ethylene glycol) diglycidyl ether, but only in a strong alkaline environment. The epoxy ring is opened during the cross-linking reaction and the resulting hydroxyl group is deprotonated. As a result, a degradation-resistant C–O–C ether bond is formed (Figure 7) [133].

Catanzano et al. [134] described chemically cross-linked chitosan using diepoxyPEG that was obtained using the solvent casting method with a 1% (*w*/*v*) solution of chitosan in 0.4% acetic acid and various amounts of diePEG of 50, 100 and 200 µL. The samples were washed to remove the unreacted diePEG. Their tests confirmed a very good stability of the final material. After 24 h of incubation in water, a loss of 2% of the initial content of the cross-linking agent in the film was observed. Moreover, the samples that contained 50 and 100 µL of the cross-linking agent saw was a slight decrease in the glass transition (Tg) of chitosan, while the sample that contained 200 µL of diePEG saw an increase in the value of the Tg. The glass transition values obtained by the authors were 135 °C, 136 °C, and 156 °C, respectively, for the samples that contained the cross-linking agent and 145 °C for chitosan. The moisture content of the films decreased with an increasing content of the cross-linking agent in the matrix, while the value of the elongation at the break parameter increased. Moreover, the authors observed that Young’s modulus decreased significantly with an increasing diePEG concentration. The degree of film swelling was reduced for the materials that contained a higher concentration of the cross-linker. The swelling value was approximately 150% after one hour of incubation in a PBS buffer for the sample that contained 200 µL. The equilibrium water content in the material decreased for the sample that contained the highest concentration of the cross-linking agent and was 66.6% compared to 85.3% for the non-cross-linked chitosan. It was found that the chemical cross-linking of chitosan did not affect the diffusion of the water vapour from the films. The value of the parameter for all of the samples was approximately 44 g/(h·m^2^).

Kiuchi et al. [107] also described the preparation of the chemical cross-linking of chitosan using diepoxyPEG. However, their studies focused on the thermal and mechanical properties. The authors used a chitosan solution with a concentration 50% lower than in the previously described study and various amounts of the cross-linking agent. These differed in their molecular weight and were dissolved in water at a range between 0.3 and 1.8 g. Based on the thermal studies, the thermal stability of the material increased as a result of cross-linking, while the molecular weight of diepoxyPEG did not influence the value of this parameter. An increase in the content of the cross-linking agent decreased the tensile strength and increased the deformation, while an increase in the molecular weight of the cross-linking agent improved the tensile strength of the hydrogel material. The chitosan films were characterised by a porous structure and the pore size was approximately 200 µm.

Sharma et al. [135] studied the chitosan hydrogels that are chemically cross-linked with bifunctional glyoxaldehyde PEG. In their study, the researchers used a 1% (*w*/*v*) chitosan solution and 6 to 24% (*w*/*v*) solutions of the cross-linking agent with a molecular weight of 2 g/mol and a content range from 3 to 12% (*w*/*v*). An increase in the value of the storage modulus was observed with an increase in the content of the cross-linking agent. The parameter value was 165.47 Pa for the sample that contained 12% (*w*/*v*) of the PEG difunctional glyoxaldehyde and 6.24 Pa for the sample that contained 3% (*w*/*v*) of the cross-linking agent. Moreover, the hydrogel exhibited self-regenerating properties. The highest degree of swelling was 120.45% and was obtained for the sample with the highest content of the cross-linking agent. However, the same sample had the slowest degradation and 45.02% of the original weight of the hydrogel was lost after ten days of incubation.

In the chemical cross-linking of chitosan, a PEG oligomer can be used as the cross-linking agent. Altinisik et al. [136] used a chitosan solution prepared in 2% (*v*/*v*) of acetic acid and a variable amount of the cross-linking agent, between 0.75 and 2.5 g. Their thermal analysis revealed two stages of film degradation, which occurred in the between 257–286 °C and 402–473 °C. An increase in the content of the cross-linking agent resulted in the formation of a fibrous structure on the surface of the material. It was also confirmed that the swelling degree of the films decreased with an increase in the PEG content in the material. The sample that contained the highest amount of the cross-linking agent degraded the slowest, and the weight loss was less than 10% after 32 days.

#### 4.1.4. 1-Ethyl-3-(3-dimethylaminopropyl) Carbodiimide and *N*-Hydroxysuccinimide (EDC/NHS)

Commonly used in the cross-linking reaction of polysaccharides and proteins, the 1-ethyl-3-(3-dimethylaminopropyl) carbodiimide (EDC) is a carbodiimide that can react with polymers that contain the carboxyl, hydroxyl, and sulfhydryl functional groups in its chemical structure group. However, it is most often used in combination with *N*-hydroxysuccinimide, which increases the reactivity of EDC by activating the less active carboxylic acids for hydrolysis. The preferred conditions for performing the cross-linking reaction are acidic or neutral, which are ensured by the presence of 4-morpholinoethanesulfonic acid (MES) or a phosphate buffer, respectively. EDC/NHS is used as a catalyst for the cross-linking reaction, which is released in the form of a substituted urea molecule. Moreover, the good solubility of both compounds in water enables the cross-linked materials to be purified by rinsing them with distilled water. The hydrogels that are obtained as a result of EDC/NHS cross-linking are characterised by a good biocompatibility, a slower degradation rate, and a greater resistance to enzymatic degradation, satisfactory mechanical properties, and a high differentiation potential for human cells. Compared to the hydrogels that are cross-linked with glutaraldehyde, this material exhibits weaker mechanical properties and a faster degradation time. Using both EDC and NHS could cause skin, respiratory, and serious eye irritations. High concentrations of these compounds are associated with the risk of cytotoxicity. Moreover, if the rinsing process is too short, the material retain unreacted EDC resulting in reaction by-products [99,103,109,137,138,139].

For example, the chemical cross-linking reaction of collagen using EDC/NHS occurs as a result of the formation of a covalent bond between the carboxyl groups of aspartic acid and glutamic acid. The cabodiimide 1-ethyl-3-(3-dimethylaminopropyl) causes the activation of the carboxylate moiety of acidic amino acid residues and the formation of an O-acyl-isourea group. The addition of NHS converts the O-acyl-isourea group into a carboxylic acid group, which can react with the amino groups. In the final stage of the reaction, EDC is converted into a substituted urea molecule (Figure 8) [113,140].

Nong et al. [141] described the preparation of collagen scaffolds cross-linked with the 1-ethyl-3-(3-dimethylaminopropyl) carbodiimide and *N*-hydroxysuccinimide in 0.033 mol/L of EDC and 0.02 mol/L of NHS, which were prepared in three different solvents: deionised water, 75% alcohol, and 95% alcohol. In order to remove the unreacted cross-linking agent, the samples were washed three times with a phosphate-buffered saline. The swelling ratio of the final materials was much lower than that of the non-cross-linked material. Moreover, the highest value of the swelling ratio (800%) was obtained for the sample that had been cross-linked with the 95% alcohol solution at 4 °C. The degree of degradation clearly decreased for the cross-linked samples compared to the non-cross-linked collagen. The highest degree of degradation was more than 10% for the sample that was cross-linked with the 95% alcohol solution at 4 °C, while the lowest value of approx. 2.5% was obtained for the sample that was cross-linked with the 75% alcohol solution at 37 °C. Based on an image of the cross-sections of the materials, the authors concluded that the cross-linking reaction positively influenced the regular distribution and the uniform pore size.

In another manuscript, Nair et al. [142] described the films cross-linked with EDC/NHS, which was used as a solution with a concentration of 10, 50, or 100%. The degree of cross-linking increased as the concentration of the cross-linking agent increased from approx. 20% for the sample that was cross-linked with the lowest concentration of the cross-linking agent to approx. 52% for the sample that was cross-linked with the 100% EDC/NHS solution. The authors also observed an increase in the value of the tensile modulus as a result of the cross-linking reaction. The tensile modulus was approximately 300 MPa for the sample that contained the cross-linking agent at the highest concentration. Moreover, the tensile stress resistance for the same sample was slightly less than 15 MPa.

Omobono et al. [143] prepared collagen hydrogels cross-linked with the aforementioned cross-linking agent. The concentration of the solution was between 10 and 100%. Their research examined the degradation process of the materials. The longest degradation time was observed for the samples that were cross-linked with the 100% and 50% cross-linker solutions as the materials did not degrade after 24 h of incubation. The shortest degradation time was obtained for the sample that was cross-linked with the 10% solution of the cross-linking agent.

Nam et al. [144] described the physicochemical properties of collagen gels cross-linked with the 1-ethyl-3-(3-dimethylaminopropyl) carbodiimide and *N*-hydroxysuccinimide. The EDC/NHS/collagen-carboxylic acid group was used at a molar ratio of 10:10:1. In order to remove any unreacted cross-linking agent, the samples were rinsed with tap water for 30 min and then immersed in 4M of NaCl for two hours. The authors concluded that the diffusion coefficient of the solvent in the gel varied with the molar concentration of the ethanol and that the highest value of the coefficient was more than 1.6·10^7^ cm^2^/min. For the optimal reaction parameters, the content of the free amino groups was 45%. The fastest rate of the cross-linking reaction was obtained for the process that lasted 24 h.

### 4.2. Natural Cross-Linking Agents

#### 4.2.1. Genipin

Genipin is a natural compound that is obtained by extracting the fruit of the Gardenia jasminoides Ellis plant. The plant comes from the regions of South Asia where it is widely used in traditional medicine. It relieves headaches and symptoms of type II diabetes and treats inflammation and liver disease. Genipin is a derivative of the non-sugar component of glycosides, i.e., the compounds that are composed of monosaccharides linked by a glycosidic bond and are used in the cross-linking of both proteins and polysaccharides. Genipin is characterised by its high selectivity because it reacts only with compounds that contain the primary amino groups in their chemical structure. Moreover, this compound exhibits biocompatibility, biodegradability, and a low cytotoxicity that is approximately 10,000 times lower than glutaraldehyde. In addition, the ability of the cells to multiply after contact with genipin is 5000 times higher than glutaraldehyde. The hydrogels that are cross-linked using genipin are characterised by a higher biocompatibility than the products that are cross-linked with their synthetic counterparts. Additionally, genipin obtains materials with increased mechanical properties and a better swelling capacity, while the degree of cross-linking of the hydrogels can be regulated by changing the pH value of the reaction medium. The degree of cross-linking significantly affects the properties of the newly-created materials such as their swelling capacity, mechanical strength, resistance to enzymatic biodegradation, and thermal stability. Currently, genipin is not widely used in mass production due ti its extremely high cost [99,100,101,102,103,145].

Collagen can be chemically cross-linked with genipin. This reaction occurs via a nucleophilic attack of the amino group on the C3 olefinic carbon atom of the cross-linking agent. As a result, the formation of a dihydropyran open ring and an unstable aldehyde group was observed. The open ring reacted with the amine group of the collagen to form a covalent bond and the intermediate aldehyde group was attacked by another amine group of the polymer to form another bond. An additional condensation reaction obtained a more stable material compared to a polymer that was cross-linked with glutaraldehyde (Figure 9) [146,147].

Tonda-Turo et al. [148] described gelatin scaffolds that were obtained by introducing genipin to an amount of 2.5 wt% (relative to the amount of the gelatin used) in a gelatin solution. The produced hydrogel was characterised by a swelling degree of 992% after 24 h. The authors observed that the cross-linking reaction of the samples increased their stability in an aqueous environment. The sponges lost 1.7% of their weight after 24 h, and 64% after two weeks. Based on the mechanical tests, the cross-linking reaction had a positive effect on the strength of the resultant materials and reduced their elastic deformability, making them stiffer. However, no influence of cross-linking on the plastic properties of the samples was demonstrated. The thermal studies confirmed that the gelatin denaturation temperature in the cross-linked gelatin was higher, which obtained a higher thermal stability of the scaffolds.

Kirchmajer et al. [149] analysed the hydrogels that were obtained using a 0 to 19.5% (*w*/*w*) concentration of a genipin solution. As a result, the degree of cross-linking of the final materials ranged from 84 to 90% and depended on the concentrations of the gelatin solution and the cross-linking agent solution. This parameter had higher values with an increasing gelatin concentration and for a decreasing genipin concentration. The authors investigated the release rate of gelatin and genipin from the cross-linked hydrogels and determined that most of the cross-linking agent, 0.89%, was released after 24 h. Moreover, the increase in the degree of cross-linking of the material limited the release of the aforementioned substances. The total time of the proteolytic degradation in the studies of Kirchmajer et al. ranged from 3.5 to 49 days, and the longest degradation time was obtained for the sample with the highest concentrations of gelatin and the cross-linking agent. Based on the mechanical studies, it was found that as the genipin concentration increased, the hydrogels became stiffer and less stretchy. However, the concentration of the cross-linking agent did not affect the value of the compressive stresses.

Yao et al. [150] obtained a gelatin material that was created using a 20% genipin solution. The final mixtures contained between 0.01 and 1.75% of the cross-linker by weight. The degree of cross-linking increased as the genipin concentration increased, and the highest degree of cross-linking was 73% for a 0.5% cross-linking agent concentration by weight. The degree of the swelling of the materials was tested in a time interval from 2.5 to 47.5 h and ranged from 280% to 620% for the samples that contained 0.05 and 0.1 wt% of genipin, and from 140% to 270% for the samples that had been cross-linked with genipin at 0.5 and 1.0 wt%. The degradation rate was also tested and the degree of the weight loss of the material that contained the lowest amount of the cross-linking agent was approximately 100% after 42 days. For the materials that contained the highest amounts of genipin, the weight loss was approx. 10% after 84 days of incubation. The glass transition temperature of the material was strictly dependent on the amount of the cross-linking agent that was used. Moreover, the sample that was cross-linked with a concentration of genipin of 0.05 wt% showed a glass transition temperature of 57 °C, while the sample that contained 0.5 wt% of the cross-linking agent was approx. 65 °C.

Bigi et al. [151] described gelatin films cross-linked using genipin with a concentration of 0.07% to 2.0% (*w*/*w*). The cross-linking process reduced the extensibility of the materials, significantly decreased the strain at break when the concentration of genipin was higher, and did not affect the value of the stress at break. The maximum degree of film cross-linking was 85% and was obtained for a genipin concentration of 1 wt%. Moreover, it was observed that when genipin was used as a cross-linking agent, it was not able to connect all of the ɛ-amino groups in gelatin. Using a high concentration of genipin prevented approx. 15% of the free ɛ-amino groups from being cross-linked. The authors observed that the degree of gelatin that was released during the incubation in the buffer solution decreased as the concentration of the cross-linking agent increased. The samples that were cross-linked with the genipin solution with a concentration of 1 wt% or more released about 2 wt% of gelatin over one month. The degree of swelling of the cross-linked material decreased with an increase in the genipin concentration. For a solution of 1 wt%, the degree of swelling was less than 200%. Moreover, an increase in the concentration of the cross-linking agent resulted in a decrease in the denaturation enthalpy value and an increase in the gelatin denaturation temperature, which positively influenced the thermal stability of the films.

#### 4.2.2. Citric Acid

Citric acid belongs to the group of hydroxycarboxylic acids and it primarily occurs in citrus fruits. Small amounts of the acid are found in living organisms that involve carbohydrate metabolism. The ionised form of citric acid is an intermediate product in the Krebs cycle, a process that enables aerobic organisms to produce energy. Citric acid is non-toxic, safe to use, and has a low cost, which is why this compound is often used in cross-linking polysaccharides and proteins. This compound plays a crucial role in regulating metabolism, preventing renal stones, regulating mineralisation, and regulating neuronal excitability in biological tissues. Ester bonds form between the polymers and citric acid as a result of its reaction with the amino and hydroxyl groups in the polymer chain. This makes it possible to improve the haemocompatibility of the hydrogel and increase the availability of the binding sites for the bioconjugation process. Additionally, studies on cellulose-based hydrogels show that cross-linking with citric acid increases the hydrophilicity of the final material, increases its surface roughness, and positively influences the differentiation of human stem cells. In addition, citric acid has a positive effect on the mechanical properties, tensile strength, and stability of the final product [22,99,103,152,153].

For example, the process of cross-linking starch that has not been modified with citric acid begins with removing the water from the structure of the cross-linking agent, thereby creating an intermediate compound(i.e., a cyclic anhydride). The ring opens under the influence of the hydroxyl groups –OH of the starch and reacts with it in the esterification reaction. In the second part of the chain, a new structure is formed with two carboxyl groups. This structure is dehydrated to form an intramolecular anhydride group. The open anhydride ring then reacts with the hydroxyl groups of the starch (Figure 10) [154].

Urang et al. [155] described gelatin materials that were obtained by mixing 30 wt% of citric acid with gelatin at 90 °C, 105 °C, and at room temperature. The highest degree of swelling and the hydrolytic degradation were obtained for the control sample at approximately 150% in the first hours of incubation and 50%, respectively. Moreover, the degree of cross-linking of the samples was dependent on the temperature during the material preparation process. The highest value of 67% was achieved at 105 °C. A higher degree of cross-linking decreased the flexibility of the films. The elongation at break decreased from approx. 17% for the control sample to less than 10% for the sample with the highest degree of cross-linking. Moreover, the materials that were subjected to the cross-linking process were characterised by better properties of the UV light barrier compared to the non-cross-linked films.

Saito et al. [156] studied gelatin gels that were chemically cross-linked with citric acid derivatives obtained by reacting citric acid and *N*-hydroxysuccinimide. The cross-linked matrix was prepared by mixing a solution of the same natural polymer as the previous research with 20 to 200 mM of the citric acid derivatives. The content of the free amino groups in the gelatin decreased as the concentration of the cross-linking agent solution increased. As a result, no free amino groups were detected in the concentrations of 50 mM or more. Moreover, the authors observed a different relationship between the degree of swelling and the concentration of the cross-linking agent. The swelling ratio of the hydrogels decreased as the concentration of the cross-linking agent increased up to 70 mM, and then gradually increased until the concentration reached 200 mM. This phenomenon was caused by the presence of residual amine groups in the matrix that were prepared with up to 70 mM of the cross-linking agent solution. An enzymatic degradation study confirmed that the hydrogel that was chemically cross-linked with citric acid derivatives at a concentration of 70 mM were characterised by the highest cross-linking density, which lost approx. 35% of its original weight after two hours of incubation.

Inoue et al. [157,158] worked on materials that were derived from alkaline gelatin cross-linked with trisuccinimidyl citrate with a final concentration of 5 to 160 mM. The samples did not gel for the cross-linking agent at a concentration of more than 80 mM. Moreover, the total enzymatic degradation of the material that was cross-linked with 20 mM of the agent took five hours. This was the maximum duration of the degradation process that was achieved. The authors’ research confirmed that the matrices were biodegradable and biocompatible, whereas the degree of swelling of the materials decreased as the concentration of the cross-linking agent increased up to 20 mM, after which the degree of swelling increased. Moreover, the cross-link density of the matrices tended to increase as the cross-linker concentration increased up to 20 mM, and then gradually decreased.

Garcia-Orue et al. [159] described hydrofilms that were based on gelatin and a citric acid solution with a concentration of 10 wt% relative to the gelatin and characterised by a swelling rate of approximately 700% after one hour of incubation. After 72 h of hydrolytic degradation, the weight of the material was 96% of the original sample weight. The final product was tested for cytotoxicity and exhibited no such effect on the cells.

#### 4.2.3. Dialdehyde Starch (DAS)

Dialdehyde starch belongs to the group of polymeric aldehydes and is obtained by the selective oxidation of starch with periodic acid or sodium periodate. The oxidation cleaves the C2–C3 bond of the anhydroglucose units that are found in native starch (unmodified starch) to form two aldehyde groups. Dialdehyde starch can be used as a cross-linking agent for polysaccharides and proteins and has starch-like properties, i.e., excellent biocompatibility and biodegradability. Moreover, it is characterised by a low toxicity and a possible antiviral activity. Studies conducted on guinea pigs and rats confirmed the lack of toxicity of the compound at a dose of 1 g/1 kg of body weight. An acute LD50 toxicity was found in oral doses equal to or greater than 6800 g/1 kg of a rat’s body weight. The use of DAS as a cross-linking agent improves the mechanical strength and thermal stability of materials, and dialdehyde starch increases the resistance to enzymatic degradation [108,113,160,161,162,163,164,165].

The reaction between the amino groups of the proteins and the formaldehyde groups of DAS occurs under mild conditions. During the cross-linking process, the structure of the protein’s triple helix does not change and their biological properties remain the same as they were before. During the process, dialdehyde starch reacts with the amino acid residues to form intermolecular bonds [166].

Cui et al. [167] described preparing gelatin hydrogels that were chemically cross-linked with the dialdehyde starch. The authors used a 10% (*w*/*v*) gelatin solution and a 10% (*w*/*v*) dialdehyde starch solution in an amount between 1 and 4 mL. The hydrogels that were produced in their experiment were characterised by a good shear strength, extensibility, and compressive strength. The compressive strength increased as the content of the cross-linking agent increased in the polymeric matrix. The maximum deformation was 80%. Moreover, the final product also had regenerative properties. The compressive stress for the sample with the highest content of the cross-linking agent was 0.55 MPa. For the same sample, the tensile strength was 5.49 kPa and the compressive strain was 1000%. Increasing the content of the cross-linking agent had a positive effect on the mechanical strength of the material. The modulus of the elasticity of the sample reached 23 kPa, which characterised the material with a high degree of elasticity. The value of the cross-linking degree of the hydrogel affected its resistance to pressure. Furthermore, an increase in the content of the dialdehyde starch reduced the size and number of pores and smoothed the material surface. Cui et al. [167] found that increasing the content of the cross-linking agent decreased the swelling speed but increased its stability.

In another manuscript, Cui et al. [168] focused on the characteristics of the cross-linking process and the examination of the textural properties of the final materials. It was observed that the gelation time of the material decreased as the content of the cross-linking agent increased. The gelation time was 22.67 s and 8.98 s for the samples that contained 1 mL and 3 mL of the cross-linking agent, respectively. For the sample that contained 4 mL of the cross-linking agent, the process was too fast for the authors to observe. The hydrogels that were produced in the reactions exhibited good injection efficiency. Moreover, the cross-linking of the materials increased their stiffness and deformation resistance. A higher concentration of the cross-linking agent resulted in the formation of a more perfect network structure and increased the damage resistance of the material. The authors characterised the textural properties of the materials as hardness, springiness, cohesiveness, gumminess, and chewiness, which increased significantly as a result of the addition of the dialdehyde starch. There was an increase in the parameter values from 326.86 g to 1268.47 g for the hardness, from 0.86 mm to 0.99 mm for the springiness, and from 0.59 N to 0.76 N for the cohesiveness. Moreover, there was an increase in the gumminess value to 944.95 mJ and the chewiness value to 638.60 mJ. Cross-linking obtained more compact and stable hydrogel material. 

Skopinska-Wisniewska et al. [169] described preparing gelatin hydrogels that were chemically cross-linked with dialdehyde starch which was 1% relative to the dry weight of the protein. The thermal results showed an increase in the temperature of the protein chain decomposition in the hydrogel from 305 °C for the non-cross-linked gelatin to 326 °C for the cross-linked sample. The process of cross-linking the material resulted in a decrease in the pore size on the surface relative to the non-cross-linked sample from 243.63 µm to 130.08 µm. For the cross-linked material, there was a significant increase in tensile strength from 31.68 kPa to 111.91 kPa, whereas the elongation value decreased slightly from 39.23 to 25.40%. The authors observed a significant increase of Young’s modulus from 37.60 kPa to 168.00 kPa, which suggested a considerable stiffness of the material. The chemical cross-linking of gelatin using dialdehyde starch positively influenced its degree of swelling—300% after one hour of incubation.

Martucci et al. [170] studied the mechanical properties of gelatin films cross-linked with dialdehyde starch in amounts between 0 and 30 wt% relative to the dry gelatin mass, which was obtained using press moulding. Using the mechanical tests, the tensile strength decreased as the content of the cross-linking agent increased from 3.9 MPa for the sample that did not contain the cross-linking agent to 1.1 MPa for the sample that contained the highest amount of dialdehyde starch. Moreover, the elongation at break for the same samples decreased from 96.9 to 55.3%. The authors found that the total soluble matter, which depended on the content of the cross-linking agent, was 100% for the non-cross-linked sample and approximately 30% for the sample that contained the highest amount of the cross-linking agent.

#### 4.2.4. Tannic Acid

Tannic acid is an organic chemical compound of a natural origin that belongs to the tannin and phenol derivatives group. It is characterised by its biocompatibility and harmlessness to the environment. Moreover, tannic acid has excellent adhesion to the skin, especially in underwater conditions. Its natural sources include, the bark of oak, chestnut, and redwood. Tannic acid has been widely used in medicine due to its anti-inflammatory, anti-microbial, and anti-cancer properties. Additionally, the compound has a haemostatic and oxidising effect. Tannic acid contains the pyrogallol and catechol groups in its structure and the molecule is composed of five double gallic acid units that are attached to a centrally located glucose unit. In addition, the hydrogels that are obtained using tannic acid have an increased resistance to enzymatic degradation due to the strong interactions between the cross-linking agent and the polymer. Moreover, tannic acid protects against the occurrence of oxidative stress in the cells that is caused by attacks of reactive oxygen species. Tannic acid can be used as a cross-linking agent in the physical and chemical cross-linking of polysaccharides and proteins, which occurs through the formation of hydrogen bonds, ionic bonds, covalent bonds, or hydrophobic interactions [104,171,172,173,174,175,176,177].

An example of using tannic acid as a cross-linking agent is a hyaluronic acid-based material that was originally cross-linked with PEGDE. The first step of cross-linking is to form covalent ether groups by reacting the hyaluronic acid hydroxyl group with the PEGDE epoxy group. In the second step, the physical cross-linking occurs between the tannic acid and the ether groups that are part of the PEGDE structure (Figure 11). The hydrogen bonds are then formed, increasing the cross-linking density of the hydrogel [104].

Azadikhah et al. [172] described preparing chitosan hydrogels that were chemically cross-linked with tannic acid in 8, 12, and 16% (*w*/*w*). The hydrogels that were obtained were characterised by a porous structure with evenly spaced pores whose size decreased as the content of the cross-linking agent increased. Based on a rheological analysis of the hydrogels, it was found that an increase in the content of the cross-linking agent increased the elasticity of the material. Moreover, the degree of the swelling of the hydrogels decreased as the content of the cross-linking agent increased, The highest value was more than 1000% after 24 h for the sample that contained 8% of the cross-linking agent. The cytotoxicity studies confirmed the good biocompatibility of the hydrogels and their low toxicity to the cells.

Michalska-Sionkowska et al. [178] studied hydrogels that were based on collagen and beta-glucan and chemically cross-linked with 2% of a tannic acid solution in the amount of 2, 5, or 10%. The authors did not observe any significant changes in the mechanical properties of the hydrogels that were dependent on the content of the cross-linking agent. The materials inhibited the dehydrogenase activity of the pathogens in a range of 6–54% depending on the bacterial strain. Moreover, the hydrogel that contained the highest amount of the cross-linking agent decreased the level of bacterial adenosine triphosphate most significantly, whereas the highest viability of human keratinocytes was observed for the sample that contained 10% of the cross-linking agent.

Ge et al. [179] described gelatin hydrogels that were cross-linked with a 0.05, 2, or 3 wt% oxidised tannic acid solution. As the oxidation of the tannic acid solution increased, the cross-linking degree of the hydrogel also increased and its mechanical properties improved. Moreover, the material was able to repair itself quickly. The authors observed that an increase in the concentration of gelatin and the cross-linking agent increased the hardness of the hydrogels up to 338.85 g for the sample that contained 20% of the gelatin solution and 3% of the cross-linking agent. For the same sample, the gumminess was 324.19 g and the resilience was 0.954. However, the highest chewiness of 954.07 g was obtained for the sample that contained 20% of the gelatin solution and the 2% of the tannic acid solution. Additionally, a springiness of 0.991 and a cohesiveness of 0.987 were obtained for the sample that contained 10% of the gelatin solution and 2% of the cross-linking agent. An increase in the oxidation state of the tannic acid resulted in an increase in the number of pores in the matrix and a decrease in their size.

Bhattacharyya et al. [180] described a gelatin hydrogel that was cross-linked with a 2.5% (*w*/*v*) tannic acid solution and used together with ferrous sulphate that was characterised by a porous structure. The degree of swelling of the hydrogels reached approx. 100% on the first day of the test, while after five days, the samples began to degrade. The authors stated that the materials had a good biocompatibility and that after three days of testing, the cell proliferation was visible on the hydrogel.

#### 4.2.5. Epigallocatechin Gallate

Epigallocatechin gallate is a compound that belongs to the group of polyphenols that is most abundant in green tea. In this raw material, the compound constitutes over 50% of the total catechin content and is characterised by the strongest antioxidant properties. Other catechins in green tea include epicatechin-3-gallate, (−)-epigallocatechin, (−)-epicatechin, and (+)-catechin. Epigallocatechin gallate has anti-inflammatory, antibacterial, and antiviral properties. In addition, there have been reports on the anti-cancer, cardioprotective, and neuroprotective effects of this catechin. Its structure is based on a combination of gallic acid and epigallocatechin. Cross-linking hydrogels using epigallocatechin gallate are mainly based on the formation of a covalent bond between the polymer and the cross-linking agent, for example using the Michael or Schiff reaction. The hydrogels that are obtained this way are characterised by their strong adhesion, self-regenerating properties, and resistance to enzymatic degradation. Moreover, the cross-linking agent positively influences the thermal and mechanical properties of the materials, and epigallocatechin gallate is characterised by a very high biocompatibility and low cytotoxicity [103,105,177,181,182,183].

For example, collagen can be chemically cross-linked with epigallocatechin gallate through a reaction between single –NH_2_ bonds, which are part of the collagen structure, and the aldehyde groups of the cross-linking agent. A stable Schiff base is formed as a result of the cross-linking reaction and the spatial structure of the collagen is not destroyed during the process, which preserves the biological properties of the polymer (Figure 12) [184].

Liu et al. [185] described preparing gelatin hydrogels that were chemically cross-linked with epigallocatechin gallate (EGCG) at a final concentration of 0.5, 1, 2, 3, or 4 g/L. The authors stated that the degree of cross-linking of the hydrogels increased with an increasing concentration of the cross-linking agent. The highest value of the degree of cross-linking was 57.1 g for the sample that contained the cross-linking agent at a concentration of 1 g/L compared to 45.7 g for the reference sample. Moreover, the thermal properties of the gelatin improved as a result of the cross-linking reaction of the material, and the highest value of the melting point and melting enthalpy was obtained for the cross-linked EGCG sample at a concentration of 1 g/L. The surface morphologies of the cross-linked and non-cross-linked samples did not differ significantly. However, increasing the concentration of the cross-linking agent resulted in the formation of pores of increasing depths.

Gou et al. [186] presented collagen that was chemically cross-linked using an epigallocatechin-3-gallate solution at a concentration of 0.1, 0.25, 0.5, 1, or 2 wt%. The highest degree of cross-linking of 21.44% was obtained for the sample that was cross-linked with the cross-linking agent at concentration of 0.5%. In addition, the cross-linking process influenced the value of the swelling coefficient. This parameter reached a value of 386.80% for the non-cross-linked sample and 701.81% for the cross-linked EGCG sample at a concentration of 0.5%, for which the best mechanical properties were obtained. The ultimate stress and elastic modulus values were 40.98 MPa and 739.04 Mpa, respectively. It was found that the cross-linking process gave directivity to the fibres on the surface of the material and caused the fibres to be more compact and thicker.

Ruan et al. [187] described films that were based on sodium alginate and carboxymethylcellulose and were cross-linked with an epigallocatechin gallate solution containing 0.4, 0.8, 1.2, or 1.6 g of the substance. The resultant films had a more compact structure compared to the samples without EGCG. The density of the material was 1.60 g/cm^3^ for the sample that contained 0.4 g of the cross-linking agent and 0.69 g/cm^3^ for the sample that did not contain the cross-linking agent. Moreover, the cross-linking process had a positive effect on the hardness and flexibility of the material. The tensile strength increased from 4.28 MPa for the sample that had not been cross-linked to 10.78 MPa for the sample that contained 1.6 g of the cross-linking agent. The elongation at break decreased from 27.50 to 11.20. Additionally, the permeability of the films decreased from 85.99 to 57.75% for the sample that contained the highest amount of the cross-linking agent, while the presence of EGCG in the matrix increased the roughness of the surface.

Li et al. [188] studied gels that were based on a myofibrillar protein that was cross-linked with epigallocatechin-3-gallate whose final content in the reaction mixture was 0.06%. The final hydrogels were characterised by a breaking force and deformation of 46.6 g and 4.7 mm, respectively, compared to 40 g and approximately 4 mm for the sample that was not cross-linked. Furthermore, the process of cross-linking the reaction mixture enabled the hardness (90.91 g), springiness (0.925 g), cohesiveness (0.666), chewiness (56.52 g), and resilience (0.395) to be increased for the cross-linked gel compared to the non-cross-linked sample. The authors observed an increase in the water holding capacity of almost 70% for the cross-linked material.

#### 4.2.6. Vanillin

Vanillin belongs to the group of phenolic aldehydes and is described as 4-hydroxy-3-methoxybenzaldehyde. This compound occurs naturally in the pods of vanilla seeds, which contain many more compounds in its structure. It has been used as a food flavouring additive, a component of fragrance compositions, and as an intermediate in cosmetics and pharmaceuticals. Naturally sourced vanillin accounts for less than 1% of its total global production. However, chemically synthesised vanillin constitutes 85% of this compound on the world market and is obtained from guaiacol. Biotechnology methods based on plants or microorganisms have also been developed, which use the biosynthetic pathways of plant tissues and cells or the biotransformation reactions of fungi, yeasts, or bacteria. Vanillin has anticancer, anti-inflammatory, antioxidant, and antimutagenic properties. The hydrogels that were obtained as a result of cross-linking with vanillin have self-healing properties due to the presence of reversible hydrogen bonds. For the same reason, the hydrogels are sensitive to temperature and pH and their stability increases at lower temperatures [189,190,191,192,193].

Cross-linking chitosan with vanillin takes place due to the reaction between the aldehyde groups of the cross-linking agent and the primary amine of the biopolymer. The result is the formation of a Schiff base (Figure 13). In addition, the hydrogen bonds between the hydroxyl groups of vanillin and the hydroxyl or the amino groups of chitosan are also formed during the cross-linking process [191,193].

Karakurt et al. [189] described films based on chitosan and polyvinyl alcohol that were crosslinked with a 1% or 3% vanillin solution. Based on the SEM micrographs, the authors observed a smooth material structure with no visible pores. The cross-linking reaction showed a stabilizing effect on the used polymer blend. The surface roughness of the non-crosslinked film was 17.9 nm, while the parameter decreased to 10.2 nm in the sample crosslinked with the 1% vanillin solution. Based on the swelling degree tests, a significant decrease in the parameter from 243.7% for the non-crosslinked material to 43.3% for the film crosslinked with a 1% solution of the crosslinking agent was observed. The authors determined the mechanical properties of the obtained samples, tensile strength, elongation at break, and Young’s modulus. Both the tensile strength and Young’s modulus increased as a result of the cross-linking agent to 59.12 MPa and 4.29 GPa, respectively. The elongation at break decreased to 1.53% for the cross-linked sample. In addition, the degree of cross-linking of the obtained materials was determined using the ninhydrin assay, which was 26.3% for the non-cross-linked sample and 69.3% for the film containing a 1% solution of the cross-linking agent.

Zou et al. [190] characterised chitosan microspheres that were chemically cross-linked with a 10% solution of vanillin and glutaraldehyde in acetone. The average particle size obtained was 19.5 µm for the glutaraldehyde-crosslinked material and 30.3 µm for the vanillin-crosslinked material. The microspheres that were cross-linked with a synthetic cross-linking agent were characterised by a more compact structure, while the samples that were cross-linked with a natural cross-linking agent had a well-defined spherical shape and a uniform surface. The authors examined the effect of the cross-linking agent on the degree of swelling of the microspheres in a PBS solution with pH 5.7. The parameter was 260% for vanillin and 250% for glutaraldehyde after 144 h of analysis. The biocompatibility studies of the week’s duration showed mild tissue inflammation for the vanillin microspheres compared to a severe inflammatory response for the glutaraldehyde sample.

Narasagoudr et al. [194] obtained films based on chitosan and polyvinyl alcohol, in which a 2% solution of ethyl vanillin in ethyl alcohol served as a cross-linking agent. For the obtained materials, a kinetic analysis was performed which determined that the average activation energy for the degradation of the cross-linked film increased to 155–213 kJ·mol^−1^ compared to the material without the cross-linking agent. In addition, the thermal analysis further confirmed the increase in the thermal stability of the film obtained as a result of the cross-linking reaction.

Da Silva et al. [191] obtained chitosan films cross-linked with a solution of vanillin at a concentration of 1%, 2%, or 3%. The largest increase in the gel content, from 57.0% to 69.8%, was observed for the sample containing a 1% solution of the cross-linking agent. In addition, based on the analysis of the mechanical properties of the film, a significant increase in Young’s modulus was observed, and the highest value of 385 MPa was obtained for the sample cross-linked with a 2% vanillin solution. The authors conducted a study of the degree of swelling. The parameter for the non-cross-linked sample was 29.4 gwater/g, while the sample cross-linked with a 3% solution of the crosslinking agent decreased to 0.92 gwater/g. Both the chitosan film without the cross-linking agent and the cross-linked film showed biocompatibility with respect to the model cells at the level of ≥90% and >78% after 48 h of exposure.

The properties of the hydrogel materials that were obtained in a chemical reaction from cross-linking biopolymers with natural and synthetic cross-linking agents are compared in Table 2.

Depending on the functional groups in the structure of the biopolymers, various types of cross-linking agents can be used in the cross-linking reaction. The summary presented in Table 3 concerns the natural and synthetic cross-linking agents that can be used in the selected chemically cross-linked biopolymers.

## 5. Biological Properties and Applications of the Hydrogels Prepared Using Natural Cross-Linking Agents

The key demands of the hydrogels for biomedical or pharmaceutical applications are biocompatibility, low or no cytotoxicity, ease of loading, and releasing drugs or their ability to be sterilised, which minimises the incidence of medical device-related infections [230]. Additionally, the biological properties of the hydrogels that are prepared using natural cross-linking agents focus on their antioxidant, antibacterial, antiviral, antifungal, and anti-inflammatory activity, which significantly determines their uses. To a large extent, polysaccharides, proteins, essential oils, and polyphenols provide these interesting properties. Moreover, some biological activity can be enhanced by using advanced biomaterials as delivery systems for antibiotics or drugs with antibacterial properties [231,232]. This is particularly important in the case of dressing materials, as the exposure of a wound site to the external environment while in moist conditions often leads to a favourable niche for bacterial growth. In turn, this phenomenon leads to inflammation, which has a key impact on preventing wound healing [233,234]. Therefore, the innate antibacterial potential and anti-inflammatory properties of the natural components offer an advantage for the improved design for wound healing applications [231,235,236,237]. On the other hand, oxidative stress has also been implicated in inhibiting the normal cell differentiation and prolonging healing and tissue regeneration [238]. Several studies have reported that the antioxidant capacity of the biomaterials consists of chitosan with phenolic compounds that is extracted from plants such as tannic acid, caffeic acid, and catechin [239,240,241]. Moreover, the antioxidant potential of the natural cross-linking agents ensures their resistance to degradation materials by the reactive oxygen species that are generated inside human tissues [104].

To date, several natural injectable hydrogels have been clinically approved by the FDA for moderate to severe facial wrinkles and folds or mid-face contour deficiencies and perioral rhytids. Some examples of these clinically injectable hyaluronic acid-based hydrogels include Teosyal^®^RHA (approved in 2017), Revanesse^®^Versa/Ultra (approved in 2017), the hyaluronic acid-based with lidocaine Belotero balance^®^ (+) Lidocaine (approved in 2019), Revanesse^®^Versa+ (approved in 2018), Restylane^®^ (approved in 2012), and the collagen-based as Evolence^®^Collagen Filler (approved in 2008) [242]. The advanced applications and future direction of hydrogels are presented in Figure 14.

## 6. Conclusions

Several synthetic cross-linking agents (*N*,*N*′-methylenebisacrylamide, ethylene glycol dimethacrylate, poly (ethylene glycol) diacrylates, epichlorohydrin, and glutaraldehyde) are commonly used to prepare hydrogels. However, this group of compounds is characterised as being harmful and toxic to both humans and the environment. The use of synthetic cross-linkers often requires additional purification steps in the technological process, which requires the input of energy, water, and solvents and, as a result, contributes to global warming. Therefore, it is worth considering the use of natural cross-linking agents such as genipin, citric acid, tannic acid, epigallocatechin gallate as well as vanillin, which produce safer hydrogels for biomedical, pharmaceutical, and cosmetic use. In this review, we focused on the chemical cross-linking of biopolymers, such as alginates, chitosan, hyaluronic acid, collagen, gelatin, and fibroin, and the use of different cross-linkers. Recent developments in the field of hydrogel production have led to the more frequent use of natural cross-linkers. This review compared and succinctly discussed the advantages and disadvantages of the synthetic and natural cross-linking agents. In the future, a new approach for synthesising chemically cross-linked hydrogels will be needed in order to reduce or eliminate certain stages of the technological processes that contribute to progressive global warming.

## Figures and Tables

**Figure 1 pharmaceutics-15-00253-f001:**
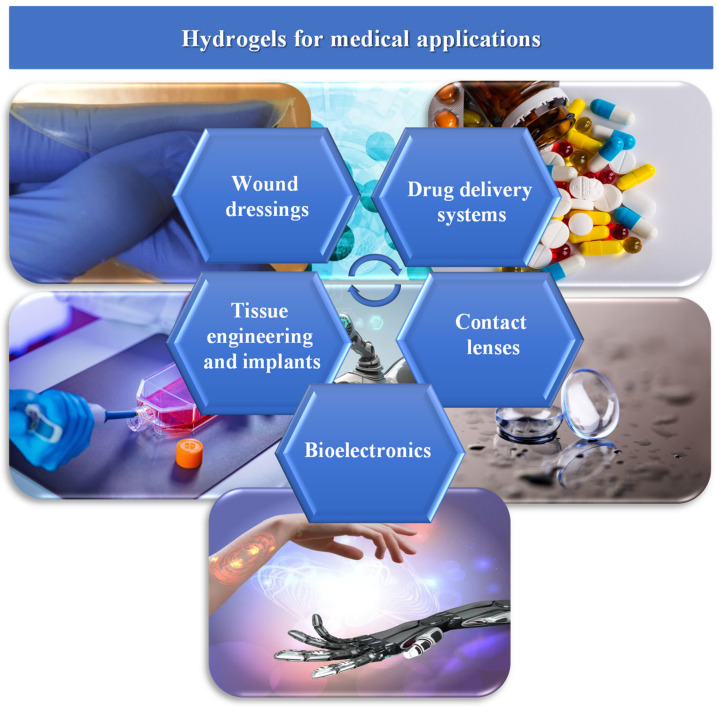
Hydrogels for medical applications.

**Figure 2 pharmaceutics-15-00253-f002:**
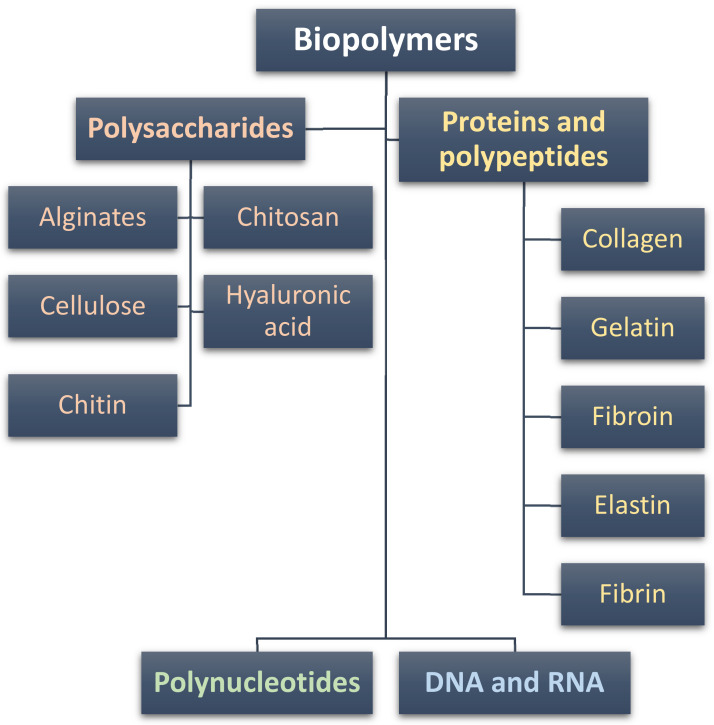
Natural biopolymers that are commonly used to prepare hydrogels.

**Figure 3 pharmaceutics-15-00253-f003:**
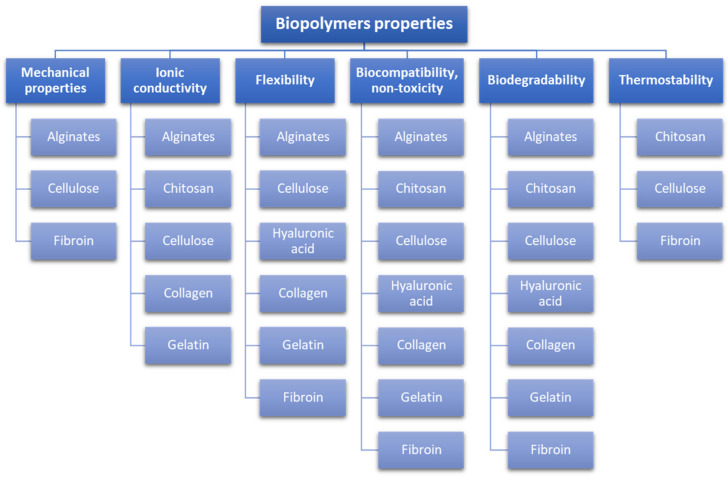
Selected properties of natural biopolymers.

**Figure 4 pharmaceutics-15-00253-f004:**
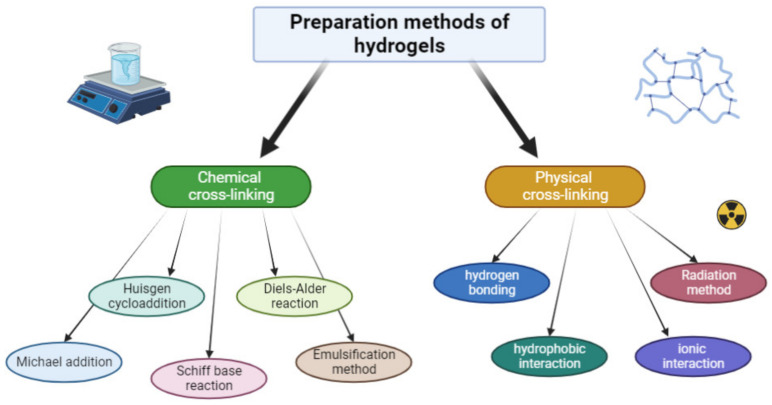
Schematic overview of the different cross-linking methods for preparing hydrogels based on biopolymers.

**Figure 5 pharmaceutics-15-00253-f005:**
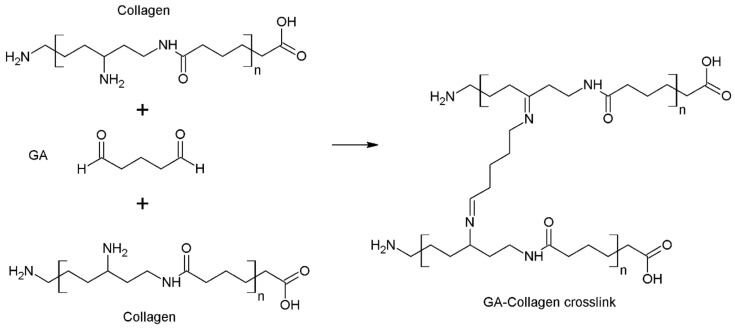
Collagen cross-linking reaction using glutaraldehyde [113,114].

**Figure 6 pharmaceutics-15-00253-f006:**
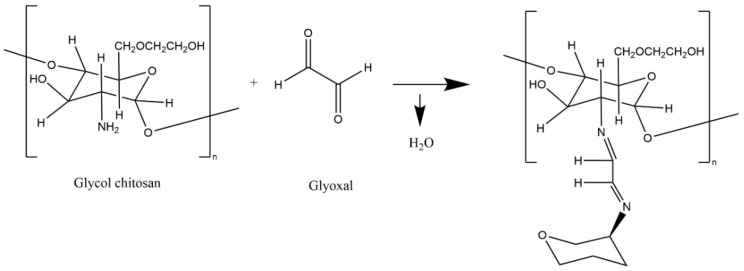
Chemical cross-linking of glycol chitosan using glyoxal [106,119,123].

**Figure 7 pharmaceutics-15-00253-f007:**
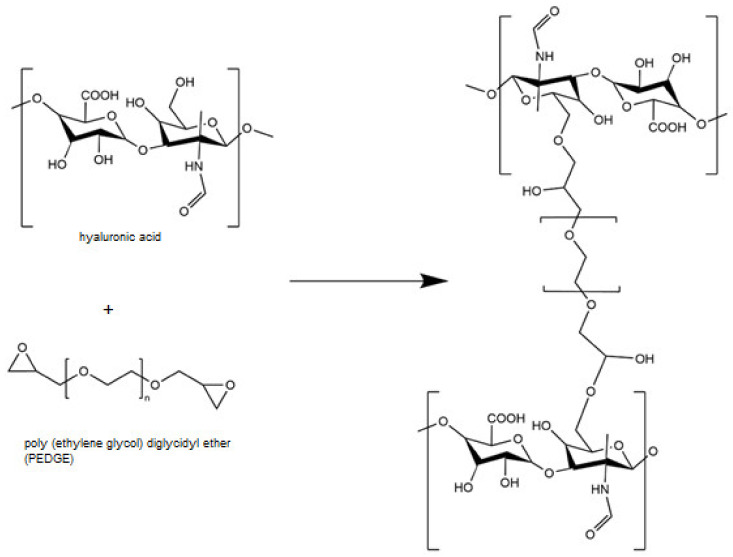
Chemical cross-linking of hyaluronic acid using PEGDE [133].

**Figure 8 pharmaceutics-15-00253-f008:**
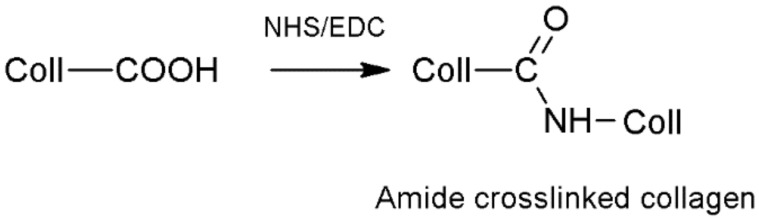
Chemical cross-linking reaction of collagen using EDC/NHS [113,140].

**Figure 9 pharmaceutics-15-00253-f009:**
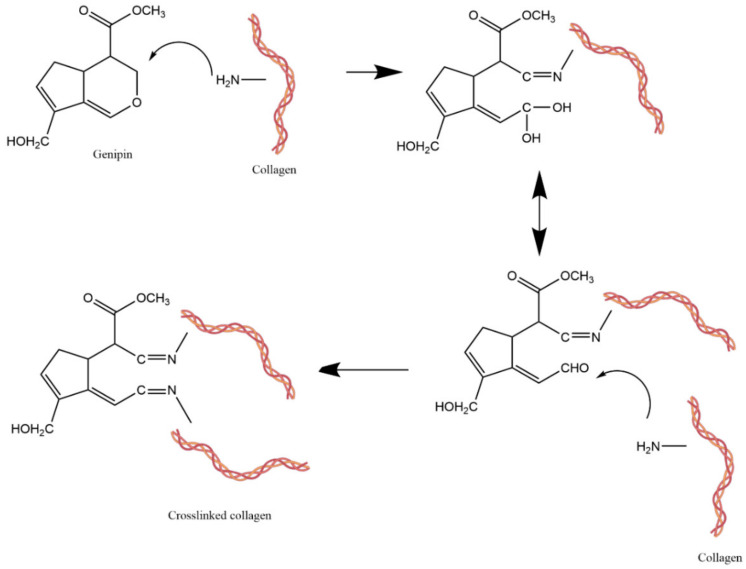
Chemical cross-linking of collagen using genipin [146,147].

**Figure 10 pharmaceutics-15-00253-f010:**
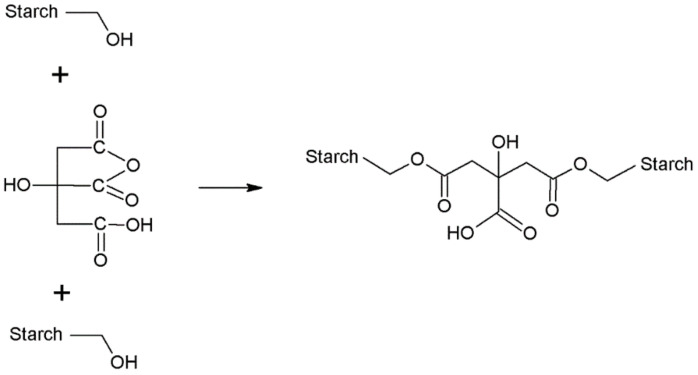
Chemical cross-linking reaction of starch using citric acid [154].

**Figure 11 pharmaceutics-15-00253-f011:**
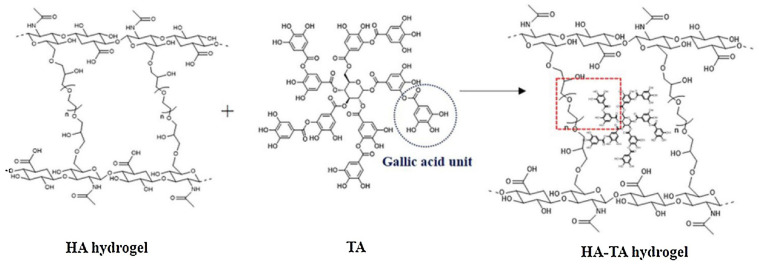
Chemical cross-linking of hyaluronic acid and PEGDE system using tannic acid [104]. Reproduced with permission from Elsevier (2022).

**Figure 12 pharmaceutics-15-00253-f012:**
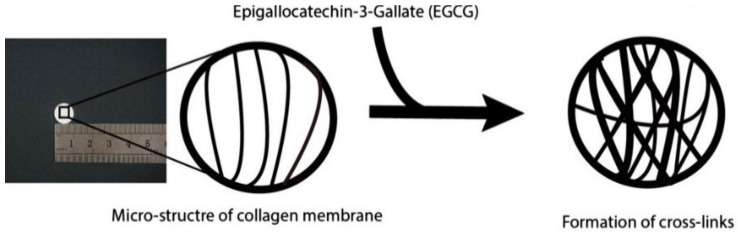
Cross-linking reaction of a collagen membrane using epigallocatechin gallate [184]. Reproduced with permission from Elsevier (2022).

**Figure 13 pharmaceutics-15-00253-f013:**
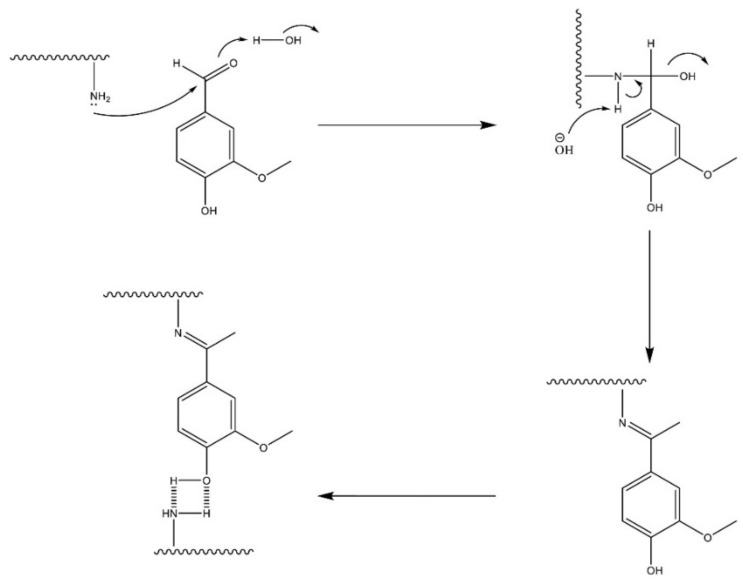
Chemical cross-linking reaction of chitosan using vanillin [193]. Reproduced with permission from Elsevier (2023).

**Figure 14 pharmaceutics-15-00253-f014:**
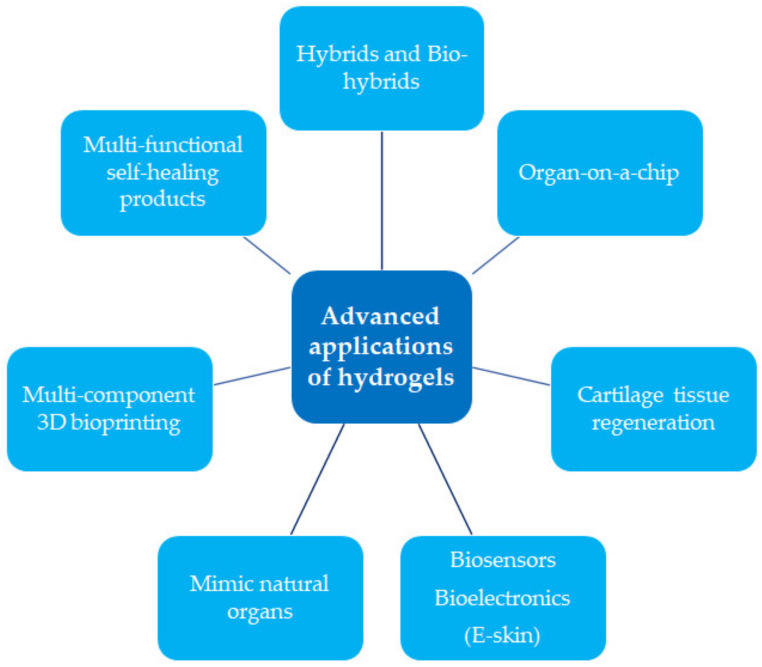
The advanced applications and future direction of hydrogels.

**Table 1 pharmaceutics-15-00253-t001:** Comparison of the properties of natural and synthetic cross-linking agents.

	Natural Cross-Linking Agent	Literature	Synthetic Cross-Linking Agent	Literature
Advantages	biocompatibility	[22,99,100,104,105]	easy availability	[99,106,107,108,109]
	biodegradability		lower production cost	
	low cytotoxicity		high reactivity	
	harmless to the environment			
	therapeutic effect			
	strong antioxidant effect			
	obtained from natural resources			
Disadvantages	higher production cost		cytotoxicity	
	lack of repeatability of the raw material		oxidative stress induction	
			harmful to the environment	
			obtained by chemical processes	

**Table 2 pharmaceutics-15-00253-t002:** Comparison of the properties of the hydrogels that were chemically cross-linked with synthetic or natural cross-linking agents.

Properties of Hydrogels	Natural Cross-Linking Agent	Synthetic Cross-Linking Agent
Cross-linking degree	−	+
Swelling degree	+	−
Degradation degree	−	+
Porosity	+	+
Hardness	−	+
Springiness	+	+
Cohesiveness	+	−
Gumminess	−	+
Chewiness	+	−
Biological safety	+	−

+ positive influence, − negative influence.

**Table 3 pharmaceutics-15-00253-t003:** Natural and synthetic cross-linking agents used in the case of selected biopolymers cross-linked chemically.

Biopolymer	Natural Cross-Linking Agent	Synthetic Cross-Linking Agent	Literature
Alginate	citric acid, oxalic acid, maleic acid, tartaric acid	glutaraldehyde, PEG-amine, adipic acid dihydrazide, EDC/NHS, epichlorohydrin, β-cyclodextrin	[195,196,197,198,199]
Cellulose	citric acid, folic acid	*N*,*N*′-methylene-bis-acrylamide, maleic acid/sodium hypophosphite, epichlorohydrin, PEGDE, glutaraldehyde	[200,201,202,203,204]
Chitosan	succinic acid, genipin, citric acid, tannic acid, tartaric acid	formaldehyde, glutaric acid, adipic acid, glutaraldehyde, glyoxal, ethylenediamine, hexamethylenediamine, acrylic acid, EDC/NHS	[205,206,207,208,209,210,211]
Collagen	epigallocatechin gallate, catechin, tannic acid, procyanidin, genipin, sodium ascorbate, oleuropein	EDC/NHS, dialdehyde starch, glutaraldehyde, 1-ethyl-3-(3-dimethylaminopropyl) carbodiimide, multi-arm PEG	[113,212,213,214,215]
Fibroin	genipin, proanthocyanidin, tannic acid	glutaraldehyde, 1-ethyl-3-(3-dimethylaminopropyl) carbodiimide, ethylene glycol diglycidyl ether, β-cyclodextrin	[60,216,217,218,219]
Gelatin	genipin, caffeic acid, tannic acid, citric acid, epigallocatechin gallate	formaldehyde, glutaraldehyde, ethylene glycol diglycidyl ether, EDC/NHS, squaric acid, dialdehyde starch	[169,220,221,222,223]
Hyaluronic acid	genipin, tannic acid, epigallocatechin gallate	PEG—dialdehyde, dimaleimide poly(ethylene glycol), EDC/NHS, carbonyldiimidazole, glutaraldehyde, butanediol-diglycidyl ether, divinyl sulfone, hydrazine sulfate	[224,225,226,227,228,229]

## Data Availability

Not applicable.

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
