# Peer review of "Are Natural Compounds a Promising Alternative to Synthetic Cross-Linking Agents in the Preparation of Hydrogels?"

_pharmaceutics, 2023, doi:10.3390/pharmaceutics15010253_

Round 1
Reviewer 1 Report
In this manuscript entitled “Are natural compounds a promising alternative to synthetic cross-linking agents in the preparation of hydrogels?”, SapuÅ‚a et al. discussed the importance and necessity of developing bio-cross-linking agents from natural resources to design functional gels. The manuscript is well written and has a wide readership. It is suitable to the scope of Pharmaceutics. I recommend to publish this paper after revised these issues.
1. For using natural polymers to develop gel materials, cellulose as the biomass is more inexpensive, abundant, and easy to process. Through the H-bonding network design, various cellulosic gel materials have been developed. The authors should add these works (Research, 2022, 2022, 9814767; Gels, 2022, 8(8): 519; Pharmaceutics 2022, 14(1), 22) in Introduction Part to demonstrate the importance of natural materials for preparing functional gels.
2. For the Figure 1, the bioelectronics such as E-skin based on hydrogels presents the great potential in medical applications, the authors should add the works and images in Figure 1.
3. Designing hydrogel materials using these biomasses such as polysaccharides, alginates, chitosan, cellulose, hyaluronic acid, proteins, collagen, gelatin, and fibroin, the molecular structures and properties determine the mechanical performances, ionic conductivity, and flexibility, the molecular structure and properties of these substances should be compared in one new Figure.
4. For the chemical cross-linking methods and process, the Figure 3 is too simple to show the linking modes, such as covalent, hydrogen, and ionic bonds.
5. Finally, the authors should give a new Figure to show the advanced applications and future direction of hydrogels.
Author Response
Manuscript ID: pharmaceutics-2125556
We would like to kindly thank Reviewer 1 for the positive comments on our manuscript. Our responses are provided below:
Comment 1: For using natural polymers to develop gel materials, cellulose as the biomass is more inexpensive, abundant, and easy to process. Through the H-bonding network design, various cellulosic gel materials have been developed. The authors should add these works (Research, 2022, 2022, 9814767; Gels, 2022, 8(8): 519; Pharmaceutics 2022, 14(1), 22) in Introduction Part to demonstrate the importance of natural materials for preparing functional gels.
Response 1: We thank the Reviewer for pointing out these references, which are now added to the introduction.
8] G. Jiang, G. Wang, Y. Zhu, A Scalable Bacterial Cellulose Ionogel for Multisensory Electronic Skin, Research. 2022 (2022) 9814767. https://doi.org/10.34133/2022/9814767.
[9] D. Filip, D. Macocinschi, M.F. Zaltariov, Hydroxypropyl Cellulose/Pluronic-Based Composite Hydrogels as Biodegradable Mucoadhesive Scaffolds for Tissue Engineering, Gels. 8 (2022) 519. https://doi.org/10.3390/gels8080519.
[10] N. Hasan, J. Lee, H.J. Ahn, Nitric Oxide-Releasing Bacterial Cellulose/Chitosan Crosslinked Hydrogels for the Treatment of Polymicrobial Wound Infections, Pharmaceutics. 14 (2022) 22. https://doi.org/10.3390/pharmaceutics14010022.
Comment 2: For the Figure 1, the bioelectronics such as E-skin based on hydrogels presents the great potential in medical applications, the authors should add the works and images in Figure 1.
Response 2: We agree with the Reviewer, that bioelectronics such as E-skin based on hydrogels are very important in medical applications and the Figure 1 has been improved.
Comment 3: Designing hydrogel materials using these biomasses such as polysaccharides, alginates, chitosan, cellulose, hyaluronic acid, proteins, collagen, gelatin, and fibroin, the molecular structures and properties determine the mechanical performances, ionic conductivity, and flexibility, the molecular structure and properties of these substances should be compared in one new Figure.
Response 3: The molecular structure and properties has been compared in new Fig 3.
Comment 4: For the chemical cross-linking methods and process, the Figure 3 is too simple to show the linking modes, such as covalent, hydrogen, and ionic bonds.
Response 4: Thank you - it has been improved. We have renumbered the figures, so now it is Fig. 4.
Comment 5: Finally, the authors should give a new Figure to show the advanced applications and future direction of hydrogels.
Response 5: Actually, the advanced applications and future direction of hydrogels are very interesting and important. We presented it in a new Figure 14.
Reviewer 2 Report
The manuscript has been well organized. The use of natural based or environmentally safe crosslinkers is important and interesting for future. So, the manuscript can be accepted for publication.
In literature review the number of references can be mentioned immediately after the names of the authors.
There are some other natural or safe crosslinkers such as vanillin and ... which acts well in film forming experiments and can be mentioned in the manuscript.
Author Response
Manuscript ID: pharmaceutics-2125556
We would like to kindly thank Reviewer 2 for the positive comments on our manuscript. Our responses are provided below:
Comment 1: In literature review the number of references can be mentioned immediately after the names of the authors.
Response 1: Thank you - it has been changed.
Comment 2: There are some other natural or safe crosslinkers such as vanillin and ... which acts well in film forming experiments and can be mentioned in the manuscript.
Response 2: It was added in the following way:
4.2.6. Vanillin
Vanillin belongs to the group of phenolic aldehydes and is described as 4-hydroxy-3-methoxybenzaldehyde. This compound occurs naturally in the pods of vanilla seeds, which contain much more compounds in its structure. It has been used as a food flavouring additive, a component of fragrance compositions and as an intermediate in cosmetics and pharmaceuticals. Naturally sourced vanillin accounts for less than 1% of its total global production. Chemically synthesised vanillin constitutes 85% of this compound on the world market and is obtained from guaiacol. Biotechnology methods based on plants or microorganisms have also been developed, which use the biosynthetic pathways of plant tissues and cells or the biotransformation reactions of fungi, yeasts or bacteria. Vanillin has anticancer, anti-inflammatory, antioxidant and antimutagenic properties. Hydrogels that were obtained as a result of cross-linking with vanillin have self-healing properties due to the presence of reversible hydrogen bonds. For the same reason, hydrogels are sensitive to temperature and pH, with stability increasing at lower temperatures [189–193].
Cross-linking of chitosan with the use of vanillin takes place as a result of the reaction of the aldehyde groups of the cross-linking agent with the primary amine of the biopolymer, the result is the formation of a Schiff base (Fig. 13). In addition, hydrogen bonds between the hydroxyl groups of vanillin and the hydroxyl or amino groups of chitosan are also formed during the cross-linking process [191,193].
Figure 13. Chemical cross-linking reaction of chitosan using vanillin [193]. Reproduced with permission from Elsevier (2022).
Karakurt et al. [189] described films based on chitosan and polyvinyl alcohol that were crosslinked with a 1% or 3% vanillin solution. Based on SEM micrographs, the authors observed a smooth material structure with no visible pores. The cross-linking reaction showed a stabilizing effect on the used polymer blend. The surface roughness of the non-crosslinked film was 17.9 nm, while the parameter decreased to 10.2 nm in the case of the sample crosslinked with 1% vanillin solution. Based on the swelling degree tests, a significant decrease in the parameter from 243.7% for the non-crosslinked material to 43.3% for the film crosslinked with a 1% solution of the crosslinking agent was observed. The authors determined the mechanical properties of the obtained samples, tensile strength, elongation at break and Young's modulus were tested. Both the tensile strength and Young's modulus increased as a result of the cross-linking agent to 59.12 MPa and 4.29 GPa, respectively. The elongation at break decreased for the cross-linked sample and amounted to 1.53%. In addition, the degree of cross-linking of the obtained materials was determined using the ninhydrin assay, which was 26.3% for the non-cross-linked sample and 69.3% for the film containing a solution of the cross-linking agent at a concentration of 1%.
Zou et al. [190] characterized chitosan microspheres chemically cross-linked with a 10% solution of vanillin and glutaraldehyde in acetone. The average particle size obtained was 19.5 µm for the glutaraldehyde-crosslinked material and 30.3 µm for the vanillin-crosslinked material. The microspheres cross-linked with a synthetic cross-linking agent were characterized by a more compact structure, while the samples cross-linked with a natural cross-linking agent had a well-defined spherical shape and a uniform surface. The authors examined the effect of the cross-linking agent on the degree of swelling of the microspheres in a PBS solution with pH 5.7, and the parameter was 260% for vanillin and 250% for glutaraldehyde after 144 hours of analysis. Biocompatibility studies of weeks duration showed mild tissue inflammation for the vanillin microspheres compared to a severe inflammatory response for the glutaraldehyde sample.
Narasagoudr et al. [194] focused on obtaining films based on chitosan and polyvinyl alcohol, in which a 2% solution of ethyl vanillin in ethyl alcohol served as a cross-linking agent. For the obtained materials, a kinetic analysis was performed, based on which it was determined that the average activation energy for the degradation of the cross-linked film increased to the value of 155-213 kJ·mol-1 compared to the material without the cross-linking agent. In addition, thermal analysis further confirmed the increase in thermal stability of the film obtained as a result of the cross-linking reaction.
da Silva et al. [191] obtained chitosan films cross-linked with a solution of vanillin at a concentration of 1%, 2% or 3%. The largest increase in the gel content, from 57.0% to 69.8%, was observed for the sample containing 1% solution of the cross-linking agent. In addition, based on the analysis of the mechanical properties of the film, a significant increase in Young's modulus was observed, and the highest value of 385 MPa was obtained for the sample cross-linked with 2% vanillin solution. The authors conducted a study of the degree of swelling, according to which the parameter for the non-crosslinked sample was 29.4 gwater/g, while for the sample crosslinked with a 3% solution of the crosslinking agent it decreased to 0.92 gwater/g. Both the chitosan film without cross-linking agent and the cross-linked film showed biocompatibility with respect to model cells at the level of ≥90% and >78% after 48 hours of exposure.
Round 2
Reviewer 1 Report
The manuscript in this state is fine.